# Creatine kinase and neuromuscular fatigue responses following differing spells of simulated cricket fast bowling

**James W. Bray**[1]*, **Chris Towlson**[1], **Stephen C. Hayes**[1], **Mark Fogarty**[2]

**1** School of Sport, Exercise and Rehabilitation Sciences, University of Hull, Kingston-Upon-Hull, United Kingdom, **2** Academic Partnership Unit, Leeds Trinity University, Leeds, United Kingdom

* j.bray@hull.ac.uk

**Data Availability Statement:** All relevant data are within the manuscript and its Supporting Information files.

## Abstract

The fast-bowling action demands repetitive high-intensity whole body movements, imposing complex physical and perceptual demands on players that vary significantly throughout the season. This study aimed to assess and establish practical methods and metrics for quantifying fatigue after four simulated fast bowling spells. Eleven senior club male fast bowlers (age 27.3 ± 7.0 y; body mass 83.7 ± 11.6 kg; height 1.80 ± 0.06 m) completed four different bouts of the modified Cricket Australia-Australian Institute of Sport bowling skills test over consecutive weeks. Neuromuscular function (countermovement jump [CMJ]) and creatine kinase (CK) levels were assessed at baseline, immediately post- (+0 h), and +24 h post-simulation. Perceptual measures (session rating of perceived exertion [sRPE]) and well-being were recorded pre and post fast-bowling simulations, with physical demands (PlayerLoad™) recorded throughout each simulation. Significant reductions in CMJ height were observed at +0 hours ($t = 9.789$, $P < 0.01$, $d = 0.50$) and +24 hours post-simulation ($t = 4.051$, $P < 0.01$, $d = 0.21$) compared to baseline. Moderate correlations were found between deliveries bowled ($r = 0.48$, $P < 0.01$), simulation duration ($r = 0.49$, $P < 0.01$), PlayerLoad™ ($r = 0.41$, $P < 0.01$), sRPE ($r = 0.48$, $P < 0.01$), and the change in CK concentration at +24 hours post-simulation. These findings suggest that lower-body neuromuscular function may be compromised following spells of fast bowing for up to 24 hours. Moreover, a 'dose-response' relationship was observed between the change in CK concentrations and PlayerLoad™, sRPE and number of deliveries bowled at +24 hours post-bowling spell. Coaches and support staff could use a combination of tools to monitor training and playing to enhance their ability to make informed decisions about a player's readiness to perform.

## Introduction

Professional cricketers experience high competition demands, which differ throughout the season as a result of multiple match formats (multi-day [$> 96$ overs.day$^{-1}$] and limited overs [50- and 20-over] cricket) [1–3]. Consequently, fast bowlers, who bowl in all formats, will experience fluctuations in demand determined by match format [4, 5], competitive strategies

**Funding:** The author(s) received no specific funding for this work.

**Competing interests:** The authors have declared that no competing interests exist.

and training [6]. Bowlers can experience both long ($\geq$ 8-overs), or short spells ($\leq$ 4-overs). This can pose a challenge for coaches and support staff as they seek to manage, maintain and/ or improve athlete performance whilst minimising injury risk [7]. Understanding the typical fatigue time-course, the physiological markers used to quantify post-match recovery (e.g., training demand, biochemical markers, and neuromuscular performance) and the 'dose-response' of fast bowling spell length (overs bowled), is important to effectively manage subsequent training and athlete readiness [2, 7–9].

Fatigue is complex and multifaceted, owing to a range of possible mechanisms [8]. Edwards [10], states that fatigue is 'a reduction in maximal or required force-generating capacity'. If left to manifest, fatigue can effect various physiological processes, such as neuromuscular function [11], markers of inflammation and muscle damage [12]. Therefore, the identification of efficient and sensitive tools to objectively quantify fatigue from both competition and training, is vital to provide valuable insights into athlete readiness [3, 13]. In applied practice, the use of a power-based movement appears preferable [14, 15]. The countermovement jump (CMJ) is the most frequently employed field test to assess lower-body neuromuscular fatigue (NMF) [14]. Furthermore, CMJ jump height can be considered an objective marker of NMF, for up to 48 h following competition [7, 16–18]. Despite widespread literature suggesting that significant NMF exists following team sport competition, current data pertaining to the NMF induced by cricket remains limited [3, 19–23]. To date, limited evidence exits explicitly describing the fatigue-related responses to spells of simulated fast bowling [19, 23], or limited overs competition [3, 22]. In both simulated fast bowling studies, negligible differences in jump height between the start and finish of spells were reported [19, 23]. Cooke, Outram [3] detailed the associated CMJ performance pre to post (including +24 h) match-play, failing to find any significant differences in jump height ($P > 0.05$). Collectively, these findings highlight that a *small* decline in NMF may exist following periods of simulated and competitive fast bowling. Nonetheless, further research is required to investigate whether there is an association between the magnitude of NMF following fast bowling spells of different lengths (i.e., 4- vs 8-overs).

Repeated high-intensity bouts of eccentric exercise, impose high mechanical strain and may result in lower limb muscle or tissue damage [9, 24–26]. The eccentric component of the fast-bowling action can cause exercise-induced muscle damage (EIMD). Blood samples can be used to analyse markers of muscle damage [14, 27]. Creatine kinase (CK), is often used as one such marker [9] and it can also be used to monitor training demand [28]. Elevated CK has been reported post competitive team sports [9, 28–30], suggesting significant muscle damage. Although a significant increase in CK has been shown to reduce ball release speed (~3%) and bowling accuracy (~79%) in fast bowlers [25] however, this was in response to resistance training and not as a result of bowling. To date, only Lombard, Muir [24] has investigated the effect of bowling spell on CK, and demonstrated that an 8-over spell was sufficient to increase indirect markers of muscle damage. However, it is important to acknowledge that this was found during the pre-season phase and the levels of EIMD and/or CK may be different, once fast bowlers have become accustomed to competition demands. Additionally, CK levels alone may not fully reflect structural muscle damage, highlighting the need to consider other indirect markers when seeking to quantify EIMD.

Bowling workload plays a crucial role in determining the likelihood of injury [31, 32]. Traditional methods of quantifying the physical demands of fast bowlers, (counting the number of balls and/or overs bowled), are prone to inaccuracies through self-reporting and assumptions of uniform intensity [33]. To enhance precision, the integration of Global Positioning System (GPS) units and inertial sensors (accelerometers [e.g., PlayerLoad™] and gyroscopes) termed micromechanical-electrical systems (MEMS), has offered coaches and support staff increased confidence in monitoring physical demands of fast-bowlers [3, 33–35]. Early studies

have used GPS to highlight player movement patterns across all formats, contributing to an enhanced understanding of fast bowling in both training [22, 36, 37] and competition [4, 5]. Recent research, has demonstrated the sensitivity and low variability ($<$7% CV) of accelerometer-derived PlayerLoad™ in detecting changes following fast bowling spells of differing durations [31]. McNamara, Gabbett [35] showed that as the prescribed bowling effort increased, the degree of variability reduced (60% effort = 19% vs. 100% effort = 7.3% CV). Furthermore, MEMS devices have demonstrated a high level of sensitivity ($>$96% accuracy) in automatically detecting fast bowling deliveries [6, 33].

Whilst existing research quantifies fast bowling demands and levels of neuromuscular and biochemical fatigue, there is insufficient evidence related to multiple bowling spells to draw a consensus of opinion. The aim of this study was therefore, to examine the 'dose-response' relationship between the length of simulated fast bowling spell and short-term neuromuscular and biochemical fatigue. It was hypothesised that longer bowling spells would lead to an increase in EIMD, influencing both NMF and CK levels. Examining measures used in applied practice will enhance the understanding of acute responses to fast bowling for practitioners enhancing fatigue management and the effectiveness of recovery practices.

## Methods

### Participants

Eleven male fast bowlers (mean ± SD; age 27.3 ± 7.0 y; body mass 83.7 ± 11.6 kg; height 1.80 ± 0.06 m) competing in senior-level club cricket, participated in this study. Data were collected over a 4-week period during the competitive season in the UK (July, 2018), with the recruitment window spanning 8-weeks (02/04/2018 to 27/05/2018). Fast bowlers were informed about the experimental procedures, along with potential risks and benefits associated with participation in the study. Informed consent was obtained in paper format and an institutional pre-exercise medical questionnaire was completed. Ethics approval was granted through the Department of Sport, Health, and Exercise Science Ethics Committee at the University of Hull (FHS:1415214). The sample size attained in this study (n = 11), is similar to those implemented in previous simulated cricket fast-bowling research [19, 23–25, 35].

### Design

This study examined the "dose-response" relationship to four different simulated fast bowling workloads by examining neuromuscular and biochemical fatigue markers, as well as perceptual responses. After a rest day, participants reported to an indoor net facility (20.9 ± 1.8˚C and 58.8 ± 11.4% relative humidity) at approximately 09:00 h. Baseline measures comprising perceptions of well-being and venous blood samples were collected. Participants then completed a standardised 10-minute warm-up, including dynamic stretches and a series of different running patterns, progressively increasing in intensity, like that previously described [3, 22, 38], followed by CMJ performance testing. Participants then completed one of four fast bowling protocols at each visit; A) 4-overs, B) 6-overs, C) Random (RAND-overs: range 36 to 60 deliveries) and D) 10-overs of simulated fast bowling, interspersed with fielding activities. After each simulation (+0 h), CMJ performance and venous blood sampling was completed. Well-being, venous blood sampling and CMJ performance were repeated 24 hours later.

### Procedures

**Neuromuscular function.** The CMJ assessment was used as an indirect measure of vertical power production. All participants performed three CMJ trials. All jumps were strictly

vertical, in that, take-off and landing were performed within the area of the jump mat (Smart Jump, Fusion Sport, Queensland, Australia) using the manufacturer software (Smart Speed, Fusion Sport, Queensland, Australia) to calculate jump height (m) and participants were instructed to perform the CMJ "as they normally would" using a self-selected depth and instructed to "jump as high as possible". The highest jump recorded for each participant was used for analysis.

**Cricket Australia-Australian Institute of Sport (CA-AIS) fast bowling skills test.** To mitigate the inherent variability in locomotive movement patterns reported during limited overs cricket match-play [4], participants completed the CA-AIS fast-bowling skills test [19]. In brief, bowlers (in pairs) performed four distinct spells of fast bowling. One player delivered a standardised set of deliveries (in a 4-over order) at match intensity, incorporating variations of line (off- and leg-stump) and length (short-, good- and full-). Meanwhile, the other participant completed the physical activity fielding simulation [19]. The same 4-over routine was repeated to extend the bowling spell, aligning with the over requirements of each experimental trial (as previously described).

**Micromechanical-electrical systems analysis.** This study used a commercially available MEMS device (MinimaxX Team Sports v2.5, Catapult Innovations, Melbourne, Australia), which contained a 100-Hz triaxial accelerometer. The device was securely fitted over the inter-scapular region of the thoracic spine, in a pocket of the manufacturer-supplied neoprene vest. A familiarisation session was completed, consisting of the CA-AIS fast-bowling skills test (used in the ensuing testing sessions) and wearing of the MEMS device. None of the participants complained of any discomfort or impediment to their performance. Data from the accelerometer embedded in the MEMS device were extracted from the commercially available software (Sprint Version 5.1.0, Catapult Innovations, Melbourne, Australia), with PlayerLoad™ calculated as the square root of the sum of the squared instantaneous rate of change in acceleration in each of the three vectors (X, Y and Z axis), divided by 100 [35, 39]. PlayerLoad™ is typically expressed as an absolute (ABS) value, with arbitrary units (AU). Relative measures of PlayerLoad™ (REL) were also calculated (ABS value [PlayerLoad™]/duration [min]; PlayerLoad™.min$^{-1}$) indicative of volume and intensity, respectively. A total of 44 files were analysed, each subsequently segmented into specific reference periods to establish bowling-only profiles. For each over examined, the footage was cropped to encompass the initial run-up for the first delivery and all subsequent movements and actions until cessation of the final delivery (six deliveries). The commencement of the initial run-up was identified by examining the raw accelerometer data, and run-up speed (total and final 5 m) was concurrently assessed using an infra-red timing system (Brower Timing Systems, Draper, USA). In addition to MEMS data, the number of balls bowled during the simulation were recorded as a measure of external demand.

**Plasma creatine kinase analysis.** Venous blood samples (6 ml) were drawn from an antecubital arm vein into a lithium heparin vacutainer (Greiner Bio-One Ltd, Stonehouse, UK). Plasma was separated by centrifugation (2300 rpm, 4°C, 10 minutes; Thermo Scientific Hereaus, Labofuge 400 R, Fisher Scientific UK, Loughborough, UK) and stored in Eppendorf tubes (Sarstedt, Numbrecht, Germany) at -80°C until analysis. Blood samples were collected at baseline (-0.5 h), immediately post (+0 h) and 24 hours post (+24 h) completion, of the modified CA-AIS fast-bowling skills test. Plasma creatine kinase (CK; U.L$^{-1}$) concentration were measured *ex vivo* using the ABX Pentra 400 auto-analyser system (Horiba, Montpellier, France). All CK analyses were conducted in duplicate.

**Perceptual responses.** Subjective perceptions of well-being were assessed using a psychological questionnaire that used a 5-point Likert scale for fatigue, sleep quality, general muscle soreness, stress levels and mood, with lower scores indicating a poorer state of well-being [22,

35, 40]. The perceived intensity of each bowling trial condition was quantified using the Borg CR-10 RPE scale [41]. Session-RPE was subsequently determined by multiplying each participants' RPE score by the duration (min) of the CA-AIS fast-bowling skills test [42]. To maintain consistent recording of ratings of perceived exertion, as outlined previously [37], sRPE scores were recorded 30 minutes after the conclusion of each trial.

### Data analyses

All statistical analysis were completed using JASP software (JASP Team 2023, Version 0.17.1). Changes in CMJ performance and muscle damage markers were analysed using repeated-measures analysis of variance (ANOVAs), and paired sample *t* tests were used to follow up any significant effects, a Bonferroni adjustment was used to reduce the chance of a type I error. Prior to the analysis, data were screened to ensure all assumptions were met. The mean difference ± 90% confidence intervals (CIs) and magnitude-based inferences were reported [43, 44], for fatigue markers and well-being at +0 and +24 hours post simulation. Threshold probabilities for a considerable effect based on 90% CIs were >0.5% most unlikely, 0.5% to 5% very unlikely, 5% to 25% unlikely, 25% to 75% possibly, 75% to 95% likely, 95% to 99.5% very likely, and >99.5% most likely. The magnitude of the observed changes were classified as: < 0.20 *trivial*, 0.20–0.59 *small*, 0.60–1.19 *moderate*, 1.20–1.99 *large* and ≥2.0 *very large*, respectively [45].

Pearson product-moment correlation (*r*) and the 95% CIs were used to assess the relationship between match demands and changes in CMJ performance, muscle-damage markers, and well-being post simulation. Differences in the simulation demands between the overs bowled were determined by using multiple 1-way ANOVAs, and Tukey post hoc used to follow up any significant effects. If heterogeneous variance was found, a variance-weighted 1-way ANOVA with the Welch method were applied. The alpha level was set at $P < 0.05$.

## Results

### Recovery

The CMJ height significantly decreased at +0 hours and +24 hours post simulation in comparison with baseline. The magnitude of change at these time points was very likely small (Table 1). Changes in CMJ height over time as a result of bowling spell duration are presented in Fig 1(A)–1(D). As a result of bowling 4-overs and of bowling 6 overs, there were significant reductions of CMJ height from baseline at +0 hours but no significant differences found +24 hours post simulation. As a result of bowling in the RAND-overs trial and the 10-overs trial there were significant reductions of CMJ height from baseline at +0 hours as well as at +24

**Table 1. Magnitude-based inferences for neuromuscular function and biochemical fatigue markers at Pre, 0- and 24-hours post simulation in comparison with baseline (mean [SD]).**

| | Pre | | 0h post | | 24h post | | Significance | | Effect Size ($\eta_p^2$) |
|---|---|---|---|---|---|---|---|---|---|
| CMJ height (m) | 0.340 | (0.054) | 0.313 | (0.052) | 0.329 | (0.053) | F(2, 86) = 48.383 | $P < 0.01$ | 0.529 |
| Creatine kinase (U.L$^{-1}$) | 436.800 | (848.723) | 543.525 | (923.323) | 551.250 | (103.773) | F(2, 78) = 1.927 | $P = 0.152$ | 0.047 |

| Post Hoc Analysis | Baseline to 0h | | | | | Baseline to 24h | | | | |
|---|---|---|---|---|---|---|---|---|---|---|
| | Mean difference ± 90% confidence interval | | *t* | Sig (*P*) | Effect size (*d*) | Qualitative Interpretation | Mean difference ± 90% confidence interval | | *t* | Sig (*P*) | Effect size (*d*) | Qualitative interpretation |
| CMJ height (m) | -0.027 | (-0.032 to -0.021) | 9.789 | < 0.01 | 0.50 | Very likely, small ↓ | -0.011 | (-0.017 to -0.005) | 4.051 | < 0.01 | 0.21 | Very likely, small ↓ |
| Creatine kinase (U.L$^{-1}$) | -106.73 | (-247.88 to 34.433) | -1.638 | 0.316 | -0.131 | Unlikely, trivial ↓ | -114.45 | (-255.61 to 26.708) | -1.757 | 0.249 | -0.140 | Very unlikely, trivial ↓ |

AU = Arbitrary units. Statistically significant difference (*P* < 0.05).

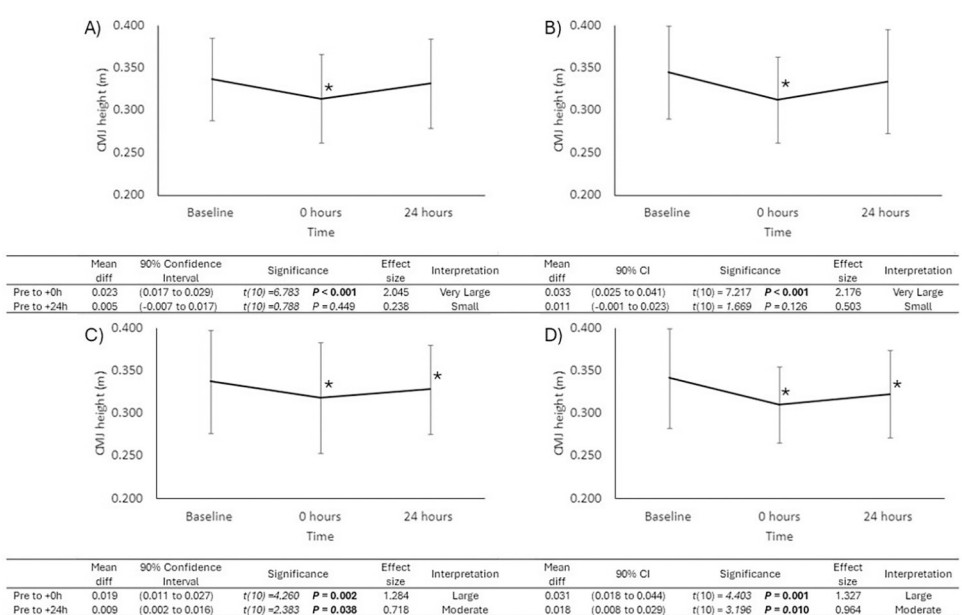

**Fig 1.** Changes in Countermovement-jump (CMJ) Height from Pre to Post (+0h and +24h) after each Simulated Fast Bowling Spell; 4-over (A), 6-over (B), RAND-over (C), and 10-over (D). *Significantly different from baseline values.

hours post simulation Additionally, significant differences in CK concentration were identified at the model level. However, CK concentration was not significantly different at +0 hours or +24 hours post simulation. The magnitude of change at these time points was unlikely to very unlikely trivial (Table 1).

Significant reductions ($F = 39.066$, $P < 0.01$) in perceived well-being were observed +24 hours post simulation. The magnitude of change at this time point was most unlikely trivial. As a result of bowling 4-overs, there was a non-significant reduction of well-being +24 hours (1.273 [90% CI, 0.073 to 2.472] AU, $t(10) = 1.923$, $P > 0.05$, $d = 0.580$ [*small*]), post simulation. In the 6-over trial, there was a significant reduction of well-being +24 hours (2.727 [90% CI, 1.455 to 3.999] AU, $t(10) = 3.886$, $P < 0.01$, $d = 1.172$ [*moderate*]), post simulation. Similarly, well-being +24 hours reduced in both the RAND-overs trial (2.273 [90% CI, 0.805 to 3.741] AU, $t(10) = 2.806$, $P = 0.02$, $d = 0.846$ [*moderate*]) and 10-overs (2.455 [90% CI, 1.325 to 3.584] AU, $t(10) = 3.938$, $P < 0.01$, $d = 1.187$ [*moderate*]) simulations, respectively.

### Relationship between physical demand and recovery

Correlations between selected simulation demands and markers of fatigue at +0 hours and +24 hours are presented in Table 2. All correlations for Δ CMJ height at +0 hours post simulation were $r < 0.3$ and are therefore not reported. Similarly, all correlations for Δ CMJ height at +24 hours post *simulation* were also $r < 0.3$ and are not reported, except for well-being, which showed a moderate and significant correlation.

All correlations for Δ CK concentration at +0 hours post simulation $r < 0.3$ and are therefore not reported, aside from sRPE, where a moderate and significant correlation was found. In the 24 hours post simulation (+24 h), a greater frequency of associations were found between the independent variables and Δ CK concentration. Typically, those external training load variables presented the largest correlations: deliveries bowled, simulation duration, PlayerLoad™ and sRPE were all moderately correlated with the Δ CK concentration at +24 hours post simulation.

**Table 2. Correlations between simulated-based bowling demands and fatigue markers at +0- and +24-hours post simulation.**

| Simulation demands | Fatigue Markers (+0 h) | | | | | | Fatigue Markers (+24 h) | | | | | |
|---|---|---|---|---|---|---|---|---|---|---|---|---|
| | Countermovement-jump | | | Creatine kinase | | | Countermovement-jump | | | Creatine kinase | | |
| | $r$ | $P$ | 95% CI | $r$ | $P$ | 95% CI | $r$ | $P$ | 95% CI | $r$ | $P$ | 95% CI |
| Deliveries bowled ($n$) | -0.04 | | -0.33 to 0.26 | 0.30 | | -0.01 to 0.55 | -0.19 | | -0.46 to 0.12 | 0.48* | | 0.20 to 0.69 |
| Duration (min) | -0.07 | | -0.36 to 0.24 | 0.28 | | -0.02 to 0.54 | -0.21 | | -0.49 to 0.09 | 0.49* | | 0.21 to 0.69 |
| Absolute PlayerLoad™ (AU) | > 0.01 | | -0.30 to 0.30 | 0.22 | | -0.09 to 0.49 | -0.11 | | -0.39 to 0.20 | 0.41* | | 0.11 to 0.64 |
| Relative PlayerLoad™ (AU.min$^{-1}$) | 0.11 | | -0.20 to 0.39 | -0.08 | | -0.38 to 0.23 | 0.19 | | -0.12 to 0.50 | -0.10 | | -0.40 to 0.22 |
| Session RPE (AU) | -0.18 | | -0.50 to 0.13 | 0.33* | | 0.03 to 0.58 | -0.28 | | -0.54 to 0.02 | 0.48* | | 0.20 to 0.69 |
| Well-being (AU) | | | | | | | 0.47* | | 0.20 to 0.67 | -0.30 | | -0.56 to 0.02 |

AU = Arbitrary units. *Significant correlation ($P < 0.05$).

## Simulation demands

Comparisons for the differences in simulation demands are presented in Table 3. ANOVA unveiled significant interactions between the number of deliveries ($F$ [3, 40] = 28.16, $P < 0.01$), final 5-m run-up speed ($F$ [3, 1846] = 6.60, $P < 0.01$), PlayerLoad™ ($F$ [3, 40] = 8.30, $P < 0.01$), sRPE ($F$ [3, 40] = 6.00, $P < 0.01$), and the number of overs bowled during each simulation, respectively. No significant interaction was found between simulation duration ($F = 2.83$, $P > 0.05$), run-up speed ($F = 1.95$, $P > 0.05$), relative PlayerLoad™ ($F = 0.34$, $P > 0.05$), baseline well-being ($F = 1.69$, $P > 0.05$), +24 hours wellbeing ($F = 2.38$, $P > 0.05$), and the number of overs bowled during each simulation. Magnitude-based inferences indicated *trivial* to *very large* effects based on the bowling simulation condition, which have been detailed in Table 3.

**Table 3. Comparison of balls bowled, duration, run-up speeds, PlayerLoad™, subjective well-being and perceived effort, mean [SD] and associated effect sizes.**

| | Four-overs (n = 11) | Six-overs (n = 11) | RAND-overs (n = 11) | Ten-overs (n = 11) |
|---|---|---|---|---|
| Deliveries (balls bowled [n]) | 24 ± 0 Six[VL], RAND[VL], Ten[VL] | 36 ± 0 RAND[VL], Ten[VL] | 48.1 ± 6 Ten[VL] | 60 ± 0 |
| Duration (min) | 34.9 ± 2.5 | 51.3 ± 6.1 | 66.7 ± 8.3 | 83.9 ± 4.2 |
| Run-up speed (km.h$^{-1}$) | 15.2 ± 2.1 | 15.3 ± 2.1 | 15.4 ± 1.9 | 15.3 ± 2.1 |
| Final 5-m run-up speed (km.h$^{-1}$) | 16.1 ± 2.5 Six[S], RAND[T], Ten[T] | 16.6 ± 2.5 RAND[T], Ten[T] | 16.4 ± 2.1 Ten[T] | 16.4 ± 2.5 |
| PlayerLoad™ (AU) | 171.3 ± 20.7 Six[VL], RAND[VL], Ten[VL] | 265.4 ± 26.2 RAND[L], Ten[VL] | 388.0 ± 35.4 Ten[VL] | 439.5 ± 56.5 |
| Relative PlayerLoad™ (AU.min$^{-1}$) | 4.9 ± 0.7 | 5.2 ± 0.7 | 5.1 ± 0.7 | 5.3 ± 0.8 |
| Perceptual well-being (AU; baseline) | 17.8 ± 1.4 | 18.3 ± 2.3 | 17.8 ± 2.0 | 18.4 ± 1.3 |
| Perceptual well-being (AU; Post [+24 h]) | 16.5 ± 2.9 Six, RAND, Ten | 15.5 ± 1.8 Four, RAND, Ten | 15.5 ± 1.8 | 15.9 ± 2.3 |
| Session RPE (AU [+0 h]) | 105.5 ± 25.9 Six[M], RAND[VL], Ten[VL] | 207.2 ± 53.7 Four[M], RAND[L], Ten[VL] | 355.5 ± 110.0 Ten[L] | 501.5 ± 142.8 |

AU = Arbitrary units. Statistically significant difference ($P < 0.05$) denoted in bold. Observed effect magnitudes are denoted as trivial ([T]), small ([S]), moderate ([M]), large ([L]), very large ([VL]).

## Discussion

The primary findings of the current study, indicate an immediate compromise in lower-body neuromuscular function after a simulated fast bowling spell, which was amplified when bowlers exceeded spells of 6-overs, and persisted for up to 24 hours post-simulation. These significant decreases in CMJ height, especially at 24 hours post, may suggest a 'dose-response' relationship between fast bowling and short-term fatigue. Despite data from other team sports showing compromised CMJ performance following match-play [40, 43, 46], to our knowledge, these findings are the first to show the presence of neuromuscular fatigue following cricket match-play or simulated fast bowling [3, 19, 22].

The only other studies that could be identified, which used CMJ height as an assessment of NMF in cricket fast bowlers were that of Cook, Outram [3] and Duffield, Carney [19]. Both of whom failed to identify differences in jump height post fast bowling spells. Various methodological differences between these studies and the current study may explain the different outcomes, including the standard of bowler used. Both Cook et al., [3] and Duffield, et al., [19] recruited first-class cricketers whereas the current study recruited senior-level club cricketers. Although the methods used in the current study followed a prescriptive method outlined in the work by Duffield, et al., [19], with bowlers alternating between bowling an over and participating in simulated fielding activities, the methods followed by Cook, et al., [3] were not sufficiently comprehensive to inform the reader of bowling and fielding activity distribution. Furthermore, it remains unclear when the recording of CMJ performance occurred in the Cooke, et al., [3] study. Consequently, it may be that professional fast bowlers are suitably conditioned to tolerate the variable demands associated with playing at this level. However, further research investigating the role of playing standard on NMF is warranted, with special consideration made to the standardisation of collection of jump data in relation to the time of bowling spell in the context of the team's innings.

Moderate correlations were found between Δ CK concentrations (24 hours post simulation) and deliveries bowled, absolute PlayerLoad™ and sRPE. Collectively, these findings are consistent with previous research reporting an increase in CK concentration following team sport match-play as a result of eccentric muscle demands [30, 43, 47] and physical contact [48]. Repeated high-intensity bouts of exercise, particularly that of the fast-bowling action, place large eccentric strain on the body and are documented as a cause of muscle damage [9, 24–26, 49]. Given that accelerometers have been shown to be sensitive in detecting differences in physical activity [39], offer a stable measure of external quantification [31], and are sensitive in the automated detection of bowling counts during training and competition [6, 33], this likely explains the similarities in correlation between both the deliveries bowled and absolute PlayerLoad™ (Table 2).

Direct comparisons of our findings to that of others was difficult given that the 'dose-response' relationship between different lengths of fast bowling and the short-term CK responses has not been previously examined, to our knowledge. Minett, et al., [21], failed to observe any significant increases in CK concentrations up to 24 h post a 10-over spell of simulated fast bowling, in agreement with our data. Although there were similarities between our protocol and that of Minett, et al., [21], their primary focus was on assessing the effectiveness of mixed-method cooling on markers of muscle damage not on the 'dose-response' to fast bowling workloads, for which cooling significantly attenuated CK in their study. In contrast to our study and the work of Minett, et al., [21], Doma, Leicht [25] reported a significant increase in CK concentrations at both 24- and 48-hours post exercise. However, Doma, et al., [25] induced muscle damage through a bowling-specific resistance training session (although the authors do not detail the specific exercises). Despite the significant increases in CK

concentrations, no change in bowling accuracy was demonstrated, post resistance training, despite changes in run-up speed and 15-m sprint time. These finding highlight the complex relationship between EIMD and performance outcomes in fast bowlers, suggesting that increases in CK concentration may not affect bowling accuracy.

As previously mentioned, the CK 'dose-response' relationship in the context of cricket fast-bowling is limited to a few studies. Research by Gastin, Hunkin [49] and Young, Hepner [50] in Australian Rules Football indicated a strong correlation between high-intensity running and the release in CK. Furthermore, Young and co-workers demonstrated a large association between PlayerLoad™ and post-match CK concentration. In addition Thorpe and Sunderland [51] identified strong relationships with high-intensity running parameters and CK in soccer. Collectively, these studies [49–51], support the notion that the repetitive, high-intensity activity could be associated with meaningful changes in CK concentrations post 24 hours. An important difference between these sports and cricket is that they typically involve some form of body contact. This may explain why, even though high-intensity fast bowling generates many of the features associated with increased CK and muscle damage no significant rises in CK were identified in the current work.

Finally, a moderate relationship between sRPE, and CK concentrations at +24 hours post simulation was identified (Table 2). This suggests that an increase in sRPE corresponds to an elevation in the magnitude of change in CK concentration following simulated fast bowling. Notably, there is limited data pertaining to the subjective demands experienced by fast bowlers and the associated short-term CK responses. However, insights from Noakes and Durandt [26], Vickery, Dascombe [37] suggest that, within the context of fast bowling, sRPE may serve as an effective tool in quantifying demands of fast bowling. Vickery, et al., [37] explored the relationship between previously established measures of training demand (e.g., # balls bowled and PlayerLoad™) and the response in sRPE. Their data identified a strong relationship between sRPE and # balls bowled and PlayerLoad™, respectively. Given the similarities between accelerometer load and CK concentration outlined earlier, we can further suggest for the use of sRPE to indirectly infer the presence of muscle damage. The simplicity, ease of interpretation, and cost-effective nature of sRPE make it a practical choice. These findings hold implications for coaches and support staff as they seek to manage fatigue, improve, and/or maintain athlete performance in the days following bouts of fast bowling.

Despite the ecological validity of this study–where players completed adaptations of a previously established fast bowling skills test [19] with spell lengths designed to mirror those in limited-overs cricket, some limitations existed. While eccentric exercise is widely recognised as a key contributor to EIMD, the CK response can vary significantly between individuals, even when factors such as gender, age and training status are accounted for. Therefore, additional indirect markers of muscle damage, particularly those with practical applications that can be easily utilised by support staff prior to competition or training (e.g., subjective questionnaires and range of movement tests), should be considered to better quantify and substantiate EIMD.

## Practical applications

Appropriate training management and monitoring of performance is now common place and is vital for those seeking to optimise training and recovery [8]. The observed decrements in CMJ confirm the sensitivity of measuring lower-body NMF in detecting fatigue following spells of fast bowling. For the first time, we have shown that length of bowling spell ($>$ 6-overs) is indicative of the magnitude of NMF experienced by fast bowlers in the days that follow. Routine monitoring of fast bowling using MEMS-derived data (i.e., PlayerLoad™) in both training and competition, is now commonplace. Consequently, coaches and support staff

should use these data to help plan training and even consider an individualised approach to recovery that accounts for the number of balls/overs bowled. The associations with accelerometer-derived load, sRPE and changes in CK concentrations following simulated fast bowling further support the idea that combination of tools to monitor workload enhances the ability to make informed decisions about a player's readiness to perform. While this technology provides comprehensive insights, the absence of such may not hinder effective monitoring. The simplicity, cost-effectiveness, and practicality of sRPE still make it a viable tool for guiding the difficult balance between increased training load and recovery for fast bowlers, ultimately preventing overtraining or injury.

## Conclusions

In conclusion, this study highlights that lower-body neuromuscular function is compromised following spells of fast bowling, which is amplified as bowling spell length increases, persisting for up to 24 hours. Furthermore, a 'dose-response' relationship was found in the change in of CK concentrations and PlayerLoad™, sRPE and number of deliveries bowled at 24 hours post bowling spell. Accordingly, subsequent recovery might vary among fast bowlers based on their individual training and playing demands, yet as bowling demand increases so does the possible need for increased recovery. These findings suggest that there is no single measure that should be used as the criterion measure of assessing fatigue, and that coaches and support staff should look to use a combination of tools to monitor training and playing, which should enhance their ability to make informed decisions about a player's readiness to perform.

## Supporting information

**S1 File. Minimal data set.**
(XLSX)

## Acknowledgments

The investigators would like to thank the participants for participation in this study.

## Author Contributions

**Conceptualization:** James W. Bray, Mark Fogarty.

**Data curation:** James W. Bray, Chris Towlson, Stephen C. Hayes.

**Formal analysis:** James W. Bray.

**Investigation:** James W. Bray, Chris Towlson, Stephen C. Hayes.

**Methodology:** James W. Bray, Mark Fogarty.

**Project administration:** James W. Bray, Chris Towlson, Stephen C. Hayes, Mark Fogarty.

**Supervision:** Mark Fogarty.

**Visualization:** Mark Fogarty.

**Writing – original draft:** James W. Bray, Chris Towlson, Mark Fogarty.

**Writing – review & editing:** Chris Towlson, Stephen C. Hayes, Mark Fogarty.

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
