## [Decision Letter · Decision Letter 0]

5 Jun 2024

PONE-D-24-16732Creatine Kinase and Neuromuscular Fatigue Responses Following Differing Spells of Simulated Cricket Fast BowlingPLOS ONE

Dear Dr. Bray,

Thank you for submitting your manuscript to PLOS ONE. After careful consideration, we feel that it has merit but does not fully meet PLOS ONE’s publication criteria as it currently stands. Therefore, we invite you to submit a revised version of the manuscript that addresses the points raised during the review process.

This is a really interesting manuscript about a under-researched topic. This, indeed, increase the interest of your paper. However, prior to publication, there are some concerns that need to be addressed. Reviewers made an detailed analysis of your paper addressing important issues to modify and there are not conflicts between reviewers. Please, take into account the metodological issues to increase the reliability and answer in a detailed way the questions about statistical tests used.

We look forward to receiving your revised manuscript.

Kind regards,

Miguel Ángel Saavedra-García, Ph.D.

Academic Editor

PLOS ONE

Journal Requirements:

Reviewers' comments:

Reviewer's Responses to Questions

**Comments to the Author**

1. Is the manuscript technically sound, and do the data support the conclusions?

Reviewer #1: Yes

Reviewer #2: Yes

2. Has the statistical analysis been performed appropriately and rigorously? 

Reviewer #1: Yes

Reviewer #2: I Don't Know

3. Have the authors made all data underlying the findings in their manuscript fully available?

Reviewer #1: Yes

Reviewer #2: Yes

4. Is the manuscript presented in an intelligible fashion and written in standard English?

Reviewer #1: Yes

Reviewer #2: Yes

5. Review Comments to the Author

Reviewer #1: Please see attached review

Reviewer #2: Introduction – The authors have done a great job highlighting the high demands placed upon fast bowlers and practitioners who look to maximise bowling performance following competition.

Methods –

[Subjects]

Line 158: Suggest remove heading “Subjects” replace with “Participants”

Line 159-160: Could the authors please provide any data demonstrating how the participants are considered “highly skilled” other than competing in senior-level club cricket? Perhaps ‘years played’, ‘training sessions per week’, ‘average peak bowling speed’, performance in ‘Cricket Australia-Australian Institute of Sport (CA-AIS) fast-bowling skills test’, if possible.

Line 163-165: Could the authors please specify whether participants were advised to avoid recovery protocols, supplementation and/or medications which may have influenced the measurements of neuromuscular and biochemical fatigue.

[Data analyses]

Under the ‘Review Questions’ section which asks, “Has the statistical analysis been performed appropriately and rigorously?” I selected “I Don’t Know” for two reasons. Firstly, I’m no expert in statistics and secondly, I’m aware of the controversy surrounding magnitude-based inference. However, as MBI has been used in conjunction with more common methods of statistical and the author making all data fully available I have no issues.

[Discussion]

Line 372-374: Suggest remove “(+0 h; - 0.027 m; 90% CI, -0.032 to -0.021; P < 0.01)” and “(-0.011 m; 90% CI, - 0.017 to -0.005; P < 0.01)” as this should have already been stated in results section. It takes away from the important result the author is trying to discuss.

Line 437-439: Could the author please address “although the authors do not detail the specific exercises”. The author mentions that Doma Leicht (25) did not detail the specific exercises used to induce muscle damage. However, I would like to clarify that Doma Leicht’s paper does in fact detail the exercises in section ‘2.6 Repetition Maximum Assessment’ & ‘2.7 Resistance Training Session’. The author could also check “Harrison, D. C., Doma, K., Rush, C., & Connor, J. (2024). Acute effects of exercise-induced muscle damage on sprint and change of direction performance: A systematic review and meta-analysis. Biology of Sport, 41(3), 153-168.” which lists the exercises, sets, reps, rest, and load. Thank you for revisiting this section of your discussion.

6. PLOS authors have the option to publish the peer review history of their article (what does this mean?). If published, this will include your full peer review and any attached files.

Reviewer #1: No

Reviewer #2: **Yes: **Drew Conor Harrison

---

## [Author Response · Author response to Decision Letter 0]

22 Jul 2024

Dear Reviewer(s),

Firstly, we would like to thank you very much for taking the time to review our original research article for consideration for publication within the ‘PLOS ONE’ journal. 

Below we address each of the individual points raised: 

Overall:

• Thank you very much for the kind words pertaining to this research area and our understanding of the literature. 

• We acknowledge the length of this paper and have worked to ensure this is clear and concise and that it now reads as a journal article versus a thesis.

Abstract:

• P2L32: grammatical error revised 

• PlayerLoad has been repositioned earlier 

Introduction:

• Thank you again for the useful and constructive feedback. Substantial sections of the introduction have been edited to condense the text ensuring a concise yet comprehensive assessment of the current literature allows the reader to understand the requirement for the study and how and why the aim and hypothesis has been developed. The two specific line examples identified in the reviewer comments have not been referenced here as the changes have been incorporated into the larger changes made to the section.

Method:

• P7L159: What makes these bowlers highly skilled? – This statement has been removed.

• Subjects has been changed to participants throughout.

• I take it your data was normally distributed as you ran an ANOVA instead of a Kruskal Wallis test? The same as a t-test instead of a Wilcoxon signed-rank test? If this is the case then it needs to be stated in your data analysis section. This information has now been included in the method section of the paper

• P10L226-233: I feel this needs to come into the methods earlier on. I was quite confused as to what the CA-AIS was until I got to this paragraph. This has been amended in the general re-organisation of the methods section.

Results:

• All tables have been re-formatted to make them easier to read and ensure that spacing is evenly distributed.

• Additionally, a substantial portion of the text has now been removed directing the reader directly to the tables which contain added details.

Discussion:

• The discussion has been edited to significantly reduce the word count and deliver a more concise appraisal of the results in the context of the current literature as suggested. As a result, the comments associated with particular lines of text have not been responded to individually as they have been edited as part of the overall process.

• Based on the comments associated with the “Practical Applications” section, no amendments have been made to this section. 

We look forward to hearing from you in due course.

Best wishes,

Dr James Bray

---

## [Decision Letter · Decision Letter 1]

1 Oct 2024

PONE-D-24-16732R1Creatine Kinase and Neuromuscular Fatigue Responses Following Differing Spells of Simulated Cricket Fast BowlingPLOS ONE

Dear Dr. Bray,

Thank you for submitting your manuscript to PLOS ONE. After careful consideration, we feel that it has merit but does not fully meet PLOS ONE’s publication criteria as it currently stands. Therefore, we invite you to submit a revised version of the manuscript that addresses the points raised during the review process. The paper still contains some fundamental flaws which I would suggest to change, as explained Reviewer 3. Although reviewer 1 has accepted his changes, as an academic editor I understand as important the contributions of reviewer 3, so I am inclined to a major revision in order to improve and adjust the quality of the paper.

In spite of everything, I consider that the article is novel and that it is well worked, well structured and well written, but the improvements indicated seem very pertinent.

We look forward to receiving your revised manuscript.

Kind regards,

Miguel Ángel Saavedra-García, Ph.D.

Academic Editor

PLOS ONE

Reviewers' comments:

Reviewer's Responses to Questions

**Comments to the Author**

1. If the authors have adequately addressed your comments raised in a previous round of review and you feel that this manuscript is now acceptable for publication, you may indicate that here to bypass the “Comments to the Author” section, enter your conflict of interest statement in the “Confidential to Editor” section, and submit your "Accept" recommendation.

Reviewer #1: All comments have been addressed

Reviewer #3: All comments have been addressed

2. Is the manuscript technically sound, and do the data support the conclusions?

Reviewer #1: Yes

Reviewer #3: No

3. Has the statistical analysis been performed appropriately and rigorously? 

Reviewer #1: Yes

Reviewer #3: Yes

4. Have the authors made all data underlying the findings in their manuscript fully available?

Reviewer #1: Yes

Reviewer #3: Yes

5. Is the manuscript presented in an intelligible fashion and written in standard English?

Reviewer #1: Yes

Reviewer #3: Yes

6. Review Comments to the Author

Reviewer #1: (No Response)

Reviewer #3: Dear Author,

Congratulations to your manuscript.

It is well-written and provides information to the practitioner working in the field of cricket.

Although, I am myself not very familiar with the sport I have some concern regarding the methodology of the paper.

In my opinion the sample size (n=11) is simply to low to draw general conlcusions. This fact is even more aggravated by the level of play of the participants, who do not present the elite population of the atletes in the sport. Therefore, the findings can not be generally accepted for high level cricketers.

CMJ jump is possibly not the best option for measuring fatigue for this sport as lower body movement contributes a much lower portion to the performance, compared to other team sports, such as football. Especilly jump height is not the most informative parameter for neuromuscular fatigue, because jumping technique can ba altered. For this please review more thouroughly the relevant literature.

ALso Player Load might not be most appropriate metric for measuring load on players. It largely correlates with distance traveled, which is obviously lower in cricket than in other team sports. Therefore, PlayerLoad might not provide the entire picture regarding the load.

Similar findings apply for the creatine kinase. Its response is highly individual specific and can be affected by many factors. However CK elevation might not be a result of the lower body action, but the arm movement, which has a much lower amount of muscle mass. In the dataset there are also obvious outliers which should be exclude, because largely affects the statistical process, thanks to the low sample size. In most cases we do not see significant increases in CK level of participants.

EVen though, the paper is carefully prepared, it still contains some fundamental flaws which I would suggest to change. First of all, a satisfying sample size with real life/match situations would be necessary to detect actual changes. Load parameters, such as PlayerLoad and CMJ metrics should be carefully chosen after thouroughly browsing the literature. Limiting factors should be minimised, if not possible, than provide the in the appropriate section.

7. PLOS authors have the option to publish the peer review history of their article (what does this mean?). If published, this will include your full peer review and any attached files.

Reviewer #1: No

Reviewer #3: No

---

## [Author Response · Author response to Decision Letter 1]

29 Nov 2024

Dear Editor and Reviewers,

Thank you for your thoughtful feedback and for providing us with the opportunity to revise our manuscript. We greatly appreciate the reviewers' constructive comments and your recognition of the manuscript’s novelty, structure, and overall quality.

We particularly acknowledge the valuable insights provided by Reviewer 3 and your recommendation for a major revision to further enhance the quality of the paper. In response, we have carefully addressed the points raised during the review process and incorporated your suggestions, as well as the key concerns outlined by Reviewer 3. A detailed rebuttal document, entitled "Response to Reviewers V2 Rebuttal (Nov)," has been uploaded for your consideration.

We deeply value your guidance and are confident that the revisions have strengthened the manuscript. Thank you again for your support and for facilitating this review process. We look forward to your review of the revised version.

Best regards,

James (+ the author team)

---

## [Decision Letter · Decision Letter 2]

3 Jan 2025

Creatine Kinase and Neuromuscular Fatigue Responses Following Differing Spells of Simulated Cricket Fast Bowling

PONE-D-24-16732R2

Dear Dr. Bray,

We’re pleased to inform you that your manuscript has been judged scientifically suitable for publication and will be formally accepted for publication once it meets all outstanding technical requirements.

Kind regards,

Miguel Ángel Saavedra-García, Ph.D.

Academic Editor

PLOS ONE

Additional Editor Comments (optional):

Reviewers' comments:

Reviewer's Responses to Questions

**Comments to the Author**

1. If the authors have adequately addressed your comments raised in a previous round of review and you feel that this manuscript is now acceptable for publication, you may indicate that here to bypass the “Comments to the Author” section, enter your conflict of interest statement in the “Confidential to Editor” section, and submit your "Accept" recommendation.

Reviewer #3: All comments have been addressed

2. Is the manuscript technically sound, and do the data support the conclusions?

Reviewer #3: Yes

3. Has the statistical analysis been performed appropriately and rigorously? 

Reviewer #3: Yes

4. Have the authors made all data underlying the findings in their manuscript fully available?

Reviewer #3: Yes

5. Is the manuscript presented in an intelligible fashion and written in standard English?

Reviewer #3: Yes

6. Review Comments to the Author

Reviewer #3: (No Response)

7. PLOS authors have the option to publish the peer review history of their article (what does this mean?). If published, this will include your full peer review and any attached files.

Reviewer #3: No

---

## [Editor Report · Acceptance letter]

10 Jan 2025

PONE-D-24-16732R2 

PLOS ONE

Dear Dr. Bray, 

I'm pleased to inform you that your manuscript has been deemed suitable for publication in PLOS ONE. Congratulations! Your manuscript is now being handed over to our production team.

Kind regards, 

on behalf of

Dr. Miguel Ángel Saavedra-García 

Academic Editor

PLOS ONE